# Mutation S115T in IMP-Type Metallo-β-Lactamases Compensates for Decreased Expression Levels Caused by Mutation S119G

**DOI:** 10.3390/biom9110724

**Published:** 2019-11-11

**Authors:** Charles J. Zhang, Mohammad Faheem, Paulie Dang, Monica N. Morris, Pooja Kumar, Peter Oelschlaeger

**Affiliations:** Department of Pharmaceutical Sciences, College of Pharmacy, Western University of Health Sciences, Pomona, CA 91766, USA; zhangcj@med.umich.edu (C.J.Z.); faheembly@gmail.com (M.F.); paulie.dang@westernu.edu (P.D.); monica.morris@westernu.edu (M.N.M.); poojakumar94@yahoo.com (P.K.)

**Keywords:** antibiotic resistance, β-lactamase, circular dichroism, thermal stability, zinc content, substrate spectrum, suppressor mutation

## Abstract

(1) Background: Metallo-β-lactamases (MBLs) have raised concerns due to their ability to inactivate carbapenems and newer generation cephalosporins and the absence of clinically available MBL inhibitors. Their genes are often transferred horizontally, and the number of MBL variants has grown exponentially, with many newer variants showing enhanced enzyme activity or stability. In this study, we investigated a closely related group of variants from the IMP family that all contain the combination of mutations S115T and S119G relative to IMP-1. (2) Methods: The effects of each individual mutation and their combination in the IMP-1 sequence background in comparison to IMP-1 were investigated. Their ability to confer resistance and their in-cell expression levels were determined. All enzymes were purified, and their secondary structure and thermal stability were determined with circular dichroism. Their Zn(II) content and kinetic constants with a panel of β-lactam antibiotics were determined. (3) Results: All four enzymes were viable and conferred resistance to all antibiotics tested except aztreonam. However, the single-mutant enzymes were slightly deficient, IMP-1S115T due to decreased enzyme activity and IMP-1-S119G due to decreased thermal stability and expression, while the double mutant did not show these defects. (4) Conclusions: These observations suggest that S119G was acquired due to its increased enzyme activity and S115T to suppress the thermal stability and expression defect introduced by S119G.

## 1. Introduction

Recent years have seen a drastic increase in the prevalence of metallo-β-lactamases (MBLs), resulting in antibiotic-resistant strains of Gram-negative bacteria, such as *Pseudomonas aeruginosa*, *Acinetobacter baumannii*, and Enterobacteriaceae, including *Klebsiella pneumoniae* [1,2]. MBLs can hydrolyze most types of β-lactam antibiotics, including new-generation cephalosporins and carbapenems, which is problematic due to the clinical importance of these “last-resort” antibiotics. No clinically available MBL inhibitors that could restore the efficacy of these drugs in the presence of MBLs exist, rendering these enzymes a significant public health issue [1,2,3]. However, there have been advances in the development of both novel β-lactam antibiotics and MBL inhibitors. Cefiderocol, a siderophore cephalosporin developed by Shionogi and Co., has activity against some strains expressing MBLs [4]. Some bicyclic boronate inhibitors under development, e.g., VNRX-5133, now known as taniborbactam (VenatoRx) [5], and QPX-7728 (Qpex Biopharma) [6] effectively inhibit MBLs.

MBLs adopt an αββα fold with the active site being located at one edge of the two central β sheets [7,8] (Figure 1a). Among the MBLs, those in the B1 subclass are the most clinically significant and include the New Delhi metallo-β-lactamase (NDM), Verona integron-borne metallo-β-lactamase (VIM), and imipenemase (IMP)-type enzymes, which share a similar active site structure. All enzymes in this subclass coordinate two Zn(II) ions. Zn1 is coordinated by three histidine residues (H116, H118, and H196; also referred to as the 3H site [9]; class B β-lactamase numbering scheme used throughout [10]), whereas Zn2 is coordinated by three different residues (D120, C221, and H263; the DCH site) (Figure 1a). The presence of both Zn(II) ions is vital to efficiently activate the β-lactam carbonyl and a hydroxide ion/water acting as the nucleophile in hydrolysis, as well as to stabilize an anionic intermediate that forms after amide bond cleavage and before protonation of the leaving nitrogen [11,12,13].

Previous studies have demonstrated the fickle nature of neighboring residues to Zn(II)-coordinating residues. For example, in NDM-type enzymes, mutations that increase the metal affinity of Zn(II)-coordinating residues greatly affect the viability and lifespan of the enzyme and are thought to be a driving factor in NDM evolution [16]. The nature of residue 262 neighboring the Zn2 ligand H263 has been shown to impact the substrate spectrum of IMP [17,18] and BcII [19] variants.

This study focuses on the active site of IMP-type MBLs and notable mutations that occur throughout several variants. IMP-14, 18, 32, 48, 49, 54, 56, 71, and 75 all contain both S115T and S119G mutations relative to IMP-1. These enzymes are closely related to each other (91.0–96.6% sequence identity) and relatively distantly related to IMP-1 (80.1–81.3% sequence identity) (Figure 1b). The variants belong to two groups: IMP-14, 32, 48, and 54 with 99.6% sequence identity between them, equaling only one mutation, and IMP-18, 49, 56, 71, and 75 with 98.8–99.6% sequence identity, equaling 1–3 mutations. No IMP variants exist with only one of the mentioned mutations, S115T or S119G; however, IMP-46, the sequence of which was published for the first time in February 2019 (GenBank Entry Code MK507819.1), has the combination of S115T and S119D. Interestingly, these mutations occur next to the Zn(II)-coordinating residues H116, H118, and D120 in loop 7, and no other mutations are found in the part of the loop facing the active site (residues 115–121) (Figure 1a). We were interested in the question of whether the co-occurrence of S115T and S119G mutations is the result of a functional role in maintaining or enhancing enzyme function or merely a coincidence. In order to explore this question, the four possible variants (no mutation, each of the two single mutations, and the double mutation) were investigated in the IMP-1 sequence background in order to isolate their effect from possible interactions with mutations elsewhere in the protein.

We found that all mutant enzymes were viable in terms of expression level, thermal stability, enzyme activity, and their ability to confer resistance. However, some distinctions could be discerned. Introduction of the S115T mutation increased thermal stability, while the S119G mutation led to relatively efficient substrate conversion, but decreased expression level and thermal stability. The combination of the two mutations led to efficient conversion of substrates, high thermal stability, good expression level, and increased minimum inhibitory concentrations (MICs) of penicillins and doripenem, suggesting that there is a functional benefit of having both mutations in combination.

## 2. Materials and Methods

### 2.1. Site-Directed Mutagenesis and Subcloning

The following plasmids were used for overexpression of MBLs: pET26b(+)-*bla*_IMP-1-S115T_ and pET26b(+)-*bla*_IMP-1-S119G_ were generated by PCR-based site-directed mutagenesis using pET26b(+)-*bla*_IMP-1_ [20] as a template. pET26b(+)-*bla*_IMP-1-S115T-S119G_ was generated by additional mutation using pET26b(+)-*bla*_IMP-1-S115T_ as the template. The following vectors were used for agar disc diffusion and MIC assays: phagemid pBC SK(+)-*bla*_IMP-1_ [20] was used as a template to create pBC SK(+)-*bla*_IMP-1-S115T_ through PCR-based site-directed mutagenesis. pBC SK(+)-*bla*_IMP-1-S119G_ and pBC SK(+)-*bla*_IMP-1-S115T-S119G_ were generated by subcloning the respective MBL genes from the pET26b(+) vectors into the empty pBC SK(+) vector. The sequences of all genes were confirmed by DNA sequencing.

### 2.2. Antibiotic Susceptibility Assays

MAX Efficiency^®^ DH10B^™^ Competent *E. coli* cells (Thermo Fisher Scientific, Waltham, MA, USA) were transformed with the pBC SK(+), pBC SK(+)-*bla*_IMP-1_, pBC SK(+)-*bla*_IMP-1-S115T_, pBC SK(+)-*bla*_IMP-1-S119G_, and pBC SK(+)-*bla*_IMP-1-S115T-S119G_ vectors by electroporation, and transformants were selected on Mueller–Hinton broth II (MHB) agar plates with 34 µg/mL chloramphenicol at 37 °C. DH10B cells without plasmid were plated on MHB agar plates without antibiotic. Multiple colonies from each plate were picked and grown in MHB containing 34 μg/mL chloramphenicol (no antibiotic for cells without vector). The cells were cultured at 37 °C for 5 h with agitation at 250 rpm. OD_600_ of each sample was determined and adjusted to 0.08 (6.4 × 10^7^ cells/mL), and these samples were used for the subsequent assays.

#### 2.2.1. Agar Disc Diffusion Assay

Sterile cotton swabs were soaked into each of the OD_600_-adjusted cultures and swabbed onto individual MHB agar plates. Antibiotic discs of ampicillin, ceftazidime, imipenem, and meropenem (Thermo Scientific Oxoid) were placed onto the agar plates. The plates were incubated for 16 h at 37 °C, and the zone of inhibition diameters were measured. The results were interpreted as susceptible, resistant, or intermediate according to the guidelines created by the Clinical and Laboratory Standards Institute (CLSI) [21]. Two-way ANOVA followed with Tukey’s multiple comparison test was used to test for significance between variants.

#### 2.2.2. Minimum Inhibitory Concentration Assay

MIC assays were done in 96 well plates. Antibiotics tested were diluted in 5% DMSO and passed through a sterile filter with 0.22 µm pores. The OD_600_ adjusted cultures were further diluted 1:100 with MHB. Ninety microliters of diluted culture were mixed with 10 µL of antibiotic in serially diluted concentrations and incubated for 20 h at 37 °C. The MIC value was determined to be the lowest concentration of antibiotic at which no visible bacterial growth could be seen. Median values of four replicates are reported. The results were interpreted as susceptible, resistant, or intermediate according to CLSI guidelines [21].

### 2.3. In-Cell Protein Expression Analysis

Expression levels of the four enzyme variants in DH10B cells harboring the corresponding pBC SK(+) vectors were determined through Western blotting. Cell cultures were adjusted to yield 1 mL with OD_600_ = 1.2, and the cultures were spun down for 3 min at 6800× *g*. The cell pellets were resuspended in 40 μL SDS-PAGE loading dye, separated on a Bio-Rad 4-15% Mini-PROTEAN TGX Precast Protein Gel, and transferred to a PVDF membrane using a Bio-Rad Mini-PROTEAN Tetra Cell. The immunostaining protocol included blocking with 5% milk in TBST (1 h, room temperature) incubation with 1:20,000 anti-IMP-1 antibody [22,23] and 1:10,000 anti-DnaK antibody (overnight, 4 °C). After washing, the membrane was cut, separating the MBL bands and the DnaK bands, incubated with 1:10,000 anti-mouse antibody-HRP conjugate and 1:10,000 protein G-HRP conjugate, respectively (1 h, room temperature), and developed with 10 mL ECL reagent (Pierce). The images were developed in the Bio-Rad ChemiDoc™ XRS+ System and analyzed densitometrically. The percentage values for the three experiments relative to IMP-1 were visualized, and statistical analysis was carried out in Prism 7 (GraphPad, San Diego, CA, USA) using one-way ANOVA followed by Dunnett’s multiple comparison test.

### 2.4. Protein Overexpression and Purification

OverExpress^TM^
*E. coli* C43 (DE3) competent cells (Lucigen, Middleton, WI, USA) were transformed by electroporation with the pET26b(+) vectors encoding the four enzyme variants and selected on lysogeny broth with 50 µg/mL kanamycin (LB + Kan) agar plates at 37 °C overnight. For each variant, one colony was picked, and an overnight culture was grown in LB + Kan at 37 °C. One liter of LB + Kan was then inoculated 1:100 with the overnight culture and incubated (37 °C with shaking at 250 rpm) until an OD_600_ of 0.7–0.9 was obtained. Isopropyl β-D-1-thiogalactopyranoside (IPTG) was added (0.5 mM final concentration) to induce enzyme expression, and the culture was further incubated at room temperature with shaking overnight. Subsequently, cells were harvested by centrifugation (3200× *g*, 4 °C, 1 h) and resuspended in 10 mL no-salt 3-(N-morpholino)propane-1-sulfonic acid MOPS buffer (50 mM MOPS, 100 µM ZnSO_4_, pH 7.0). Resuspended bacteria were then lysed through sonication while placed on ice. Cell debris was removed by centrifugation (30,000× *g*, 4 °C, 45 min), and the supernatant was passed through a sterile filter with 0.22 µm pores. The filtrate was injected into three connected HiTrap CM FF ion-exchange columns (GE Life Sciences), which were then connected to an ÄKTA Purifier UPC 100 with Fraction Collector (GE Life Sciences). After washing with four column volumes of no-salt MOPS buffer, a salt gradient was run by gradually increasing the amount of high-salt MOPS (50 mM MOPS, 100 µM ZnSO_4_, 1 M NaCl, pH 7.0). Fractions collected while peaks of absorbance at 280 nm were detected were tested for MBL presence through mixing an aliquot of the fraction with chromacef. Fractions positive for β-lactamase activity (color change to red) were combined and concentrated by centrifuging in a 15 mL Ultra Centrifugal Filter Unit with a molecular weight cut-off of 10 kDa (Amicon) for 15 min at 3200× *g*, 4 °C. Proteins were then further separated and excess salt removed using size-exclusion chromatography on a HiPrep^TM^ 26/60 Sephacryl^TM^ S-100 HR column (GE Life Sciences) and low-salt MOPS buffer (50 mM MOPS, 100 µM ZnSO_4_, 100 mM NaCl, pH 7.0). Again, protein was monitored at 280 nm and enzyme activity with chromacef, and all fractions positive for MBL activity were pooled and concentrated. The final product was separated into 20 µL aliquots and stored at −20 °C for further experiments. The protein purity was ascertained by SDS-PAGE and Coomassie staining, and protein concentration was determined spectrophotometrically using a molar extinction coefficient of 49,000 M^−1^ cm^−1^ at 280 nm previously determined for IMP-1 [24].

### 2.5. Biophysical Characterization of Enzyme Variants

#### 2.5.1. Mass Spectrometry

Samples of purified MBLs were sent to Dr. Mona Shahgholi’s Mass Spectrometry Laboratory at the California Institute of Technology (Pasadena, CA, USA) for liquid chromatography-mass spectrometry with electrospray ionization. The experimentally determined masses were compared to theoretical masses calculated by entering the relevant amino acid sequences into the Compute pI/Mw Tool (http://web.expasy.org/compute_pi/) [25].

#### 2.5.2. Determination of Zn(II) Content

Samples of the purified enzymes were dialyzed 1:8,000 against Zn(II)-free MOPS buffer (50 mM MOPS, pH 7.0) in FloatALyzer dialysis tubes with an 8–10 kDa molecular weight cut-off (Spectrum, Gardena, CA, USA) for 21 h at 4 °C. Protein concentration after dialysis was determined as described above. The protein was diluted to 1 µM using 4-(2-pyridylazo)resorcinol (PAR) assay solution (4.52 M GdnHCl, 97 µM PAR, 50 mM MOPS, pH 7.0) [18,26]. Zn(II) standards were prepared from PAR assay solution containing a ZnSO_4_ concentration ranging from 0 to 4 µM. All samples were incubated overnight at room temperature, and absorbance at 500 nm was measured for all IMP samples and Zn(II) standards using an SQ-2802 UV/Vis Spectrophotometer (UNICO, Dayton, NJ, USA). A_500_ values of the Zn(II) standards were plotted against Zn(II) concentration, and a linear trendline equation was obtained, which was then used to determine Zn(II) content of the IMP samples. The assay was done in triplicate.

#### 2.5.3. Circular Dichroism Experiments

Samples were dialyzed against PBS buffer (50 mM PBS, 100 μM ZnSO4, pH 7.0). All samples were diluted to 300 µL of 5 µM IMP enzyme. The entire diluted sample was then loaded into a cuvette with a 0.1 cm path length. For every IMP sample three CD scans were done using a J-715 Spectropolarimeter (JASCO, Easton, MD, USA) between 250 nm and 190 nm and averaged to receive one spectrum. The values given by the instrument were in millidegrees and indicated the ellipticity of the sample. The ellipticity was then converted to ellipticity per residue (θ) using the following Equation (1):(1)$$\theta \;=\;\left(\mathrm{CD}\;\mathrm{signal},\;m\;\deg\right)\left(\frac{\deg}{10^{3}m\;\deg}\right)\left(\frac{1}{pathlength\;\mathrm{cm}}\right)\left(\frac{1\;L}{protein\;µmol}\right)\left(\frac{10^{3}}{1\;L}\right)\left(\frac{1\mathrm{cm}3}{1\;mL}\right)\left(\frac{10^{6}µmol}{1\;mol}\right)\left(\frac{1mol}{10\;dmol}\right)\left(\frac{1}{resid}\right)$$

Using DichroWeb [27,28] and the K2D program, percentages of secondary structure components were calculated from the four CD spectra.

To monitor thermal denaturation of the enzymes, CD ellipticity was constantly recorded at 215 nm while the samples were heated from 25 °C to 100 °C and plotted against temperature. To normalize the data, linear equations were calculated for the pre-transition extension and the post-transition extension, at which the proteins exist in their native and unfolded states, respectively. The following equation was then used to obtain the fraction unfolded:
(2)$$U=\frac{\left(\left[\theta \right]-\left(\theta_{N}^{o}\;+m_{N}\left[T\right]\right)\right)}{(\left(\theta_{U}^{o}\;+m_{U}\left[T\right]\right)-\;\left(\theta_{N}^{o}\;+m_{N}\left[T\right]\right)}$$ where *U* indicates the fraction unfolded, *θ* the ellipticity measured, T the temperature in K, $\theta_{N}^{o}$ and *m_N_* the y-intercept and slope of the pre-transition extension, respectively (N for native), and $\theta_{U}^{o}$ and *m_U_* the y-intercept and slope of the post-transition extension, respectively (U for unfolded). The thermal denaturation curves were finally fitted using a non-linear function in Prism 7 to determine melting temperatures.

### 2.6. Enzyme Kinetic Assays

Steady-state kinetic experiments were carried out with the four IMP variants and the antibiotics benzylpenicillin, ampicillin, cephalothin, cefoxitin, cefotaxime, ceftazidime, imipenem, meropenem, doripenem, and aztreonam. To monitor the hydrolysis of substrates, absorbance was recorded with an SQ-2802 UV/Vis Spectrophotometer equipped with an AS-21P Peltier/Sipper Controle System (UNICO) set to 30 °C. Purified IMP variants were prepared in 50 mM MOPS buffer (pH 7.0) supplemented with 100 μM ZnSO_4_ and 10 μg/mL bovine serum albumin (BSA) to prevent MBL denaturation at low concentrations. The substrates were prepared in the same buffer without BSA. Both solutions were preincubated at 30 °C for 5 min. The enzyme solution was added into a quartz cuvette first, equilibrated and blanked, and the substrate solution was added, mixed, and absorbance at the wavelengths corresponding to the β-lactam amide bond recorded for two minutes at half second intervals. The specific wavelengths and molar extinction coefficients of all substrates have been reported previously [29]. For each substrate, eight concentrations were used, and three series of experiments were carried out. Nonlinear fits to the Michaelis–Menten equation were obtained with Prism 7 for each series, and the kinetic constants reported are the means ± standard deviations.

## 3. Results

### 3.1. Site-Directed Mutagenesis and Subcloning

Mutations were introduced into the pET26b(+)-*bla*_IMP-1_ plasmid using PCR-based site-directed mutagenesis to create pET26b(+)-*bla*_IMP-1-S115T_ and pET26b(+)-*bla*_IMP-1-S119G_. pET26b(+)-*bla*_IMP-1-S115T_ was subsequently used as a template to introduce the second mutation to obtain pET26b(+)-*bla*_IMP-1-S115T-S119G_. All mutations were confirmed by DNA sequencing. The genes were then subcloned into the pBC SK(+) phagemid vectors or generated by PCR-based site-directed mutagenesis from pBC SK(+)-*bla*_IMP-1_. Electrocompetent *Escherichia coli* OverExpress C43 cells (Lucigen, Middleton, WI, USA) were transformed with the different pET26b(+) plasmids for protein overexpression and purification. Electrocompetent *E. coli* DH10B cells (Thermo Fisher Scientific, Waltham, MA, USA) were transformed with the pBC SK(+) plasmids to be used for agar disc diffusion and minimum inhibitory concentration (MIC) assays, as well as in-cell expression analysis.

### 3.2. Antibiotic Susceptibility Assays

#### 3.2.1. Agar Disc Diffusion Assay

The agar disc diffusion assay revealed no difference between *E. coli* DH10B cells expressing the four different enzyme variants from the pBC SK(+) vector. All cells expressing MBLs were resistant against the penicillin ampicillin and the third-generation cephalosporin ceftazidime (zone of inhibition diameter of 0 mm), while the negative controls (cells not harboring the vector or an empty vector) were susceptible (Figure 2a and Appendix A). Zone of inhibition diameters for both carbapenems (imipenem and meropenem) were similar between the four enzyme variants and significantly smaller than those for the negative controls; however, according to CLSI breakpoints [21], cells expressing the enzymes were still susceptible to imipenem and meropenem.

#### 3.2.2. Minimum Inhibitory Concentration Assay

A different trend can be seen from the MIC values (Figure 2b and Table 1). All four IMP variants conferred resistance to every bicyclic β-lactam tested, but not to the monocyclic aztreonam, based on CLSI breakpoints [21]. Overall, the differences between the four variants were not dramatic (no more than eight-fold or three serial dilutions within each antibiotic). The MICs of cefoxitin and imipenem were identical for all four variants and those of cephalothin, cefotaxime, and aztreonam within one serial dilution. There appeared to be a consistent pattern with the penicillins (benzylpenicillin and ampicillin), the zwitterionic third-generation cephalosporin ceftazidime, and the two newer carbapenems (meropenem and doripenem): the MICs were always higher or the same with IMP-1 and the double mutant IMP-1-S115T-S119G than with the single mutants IMP-1-S115T and IMP-1-S119G. The double mutant conferred higher resistance levels than IMP-1 to both penicillins and doripenem, although the differences were only one serial dilution.

### 3.3. In-Cell Expression Levels

Western blotting was employed using the same cells and vectors as for antibiotic susceptibility assays (*E. coli* DH10B and pBC SK(+)) to compare enzyme expression levels between the four IMP variants (Figure 3 and Table 2). The relative expression levels of IMP-1-S115T and IMP-1-S115T-S119G did not deviate significantly from that of IMP-1. However, the expression level of IMP-1-S119G was only half of that of IMP-1, which is statistically significant (*p* < 0.01).

### 3.4. Enzyme Overexpression, Purification, and Biophysical Characterization

#### 3.4.1. Enzyme Overexpression and Purification

One liter cultures of *E. coli* OverExpress C43 (DE3) cells transformed with pET26b(+) vectors encoding the different *bla*_IMP_ genes were grown at 37 °C until an OD_600_ of 0.7–0.9 was obtained, induced with 0.5 mM IPTG and incubated at room temperature overnight. Cells were harvested by centrifugation, lysed by sonication, and the resulting soluble fraction submitted to cation exchange chromatography followed by size exclusion chromatography. Finally, purified protein solutions in 50 mM MOPS, 100 µM ZnSO4, and 100 mM NaCl, pH 7.0 were concentrated to a small volume (1–2 mL), and the protein concentration was determined by measuring absorbance at 280 nm. All enzyme preparations were at least 95% pure as judged by SDS-PAGE. The final concentrations and overall yields obtained from the 1 L culture are summarized in Table 2.

#### 3.4.2. Mass Spectrometry

The molecular masses of the different enzyme variants determined by electrospray ionization mass spectrometry were in good agreement with those calculated based on the amino acid sequences and using the Compute pI/Mw program [25] (Table 2).

#### 3.4.3. Zn(II) Content

The PAR assay [26] adapted to our buffers [18] was used to determine the number of Zn(II) ions bound per enzyme molecule and revealed that all IMP variants contained approximately two equivalents of Zn(II) ions (Table 2).

#### 3.4.4. Circular Dichroism Experiments

CD experiments were employed to, firstly, ascertain that the secondary structure of IMP-1 was not compromised by introducing mutations and, secondly, examine the impact of mutations on thermal stability. CD scans from 250 to 190 nm of the four enzymes were nearly superimposable, and the differences in molar ellipticities were likely due to errors in protein concentration determination (Figure 4a). Analysis with the program DichroWeb [27,28] suggests that the secondary structures of the proteins are equivalent. Percentages of different components are shown in Table 2.

Thermal denaturation was examined by the disappearance of the CD signal at 215 nm upon continuous heating of the samples. Melting curves are shown in Figure 4b and melting temperatures in Table 2. They indicate a slight destabilization of IMP-S119G, but stabilization of IMP-1-S115T and IMP-1-S115G-S119G, relative to IMP-1. Under the assay conditions (30 °C for enzyme activity assays and 37 °C for susceptibility assays), all enzymes were expected to exist 100% in their native conformation.

### 3.5. Enzyme Kinetic Constants

Kinetic constants of the four purified enzymes with a range of β-lactam antibiotics representing penicillins, cephalosporins, a cephamycin, carbapenems, and the monobactam aztreonam (identical to those tested in MIC assays) were determined and are shown in detail in the Appendix A (Appendix A). Depending on substrate concentration achieved in the bacterial periplasm and the enzymes’ affinity to each substrate, either the turnover number *k*_cat_ (high substrate concentration, low *K*_M_) or catalytic efficiency *k*_cat_/*K*_M_ (low substrate concentration, high *K*_M_) will determine resistance levels conferred by the enzymes. Therefore, *k*_cat_ and *k*_cat_/*K*_M_ are presented in Figure 5.

For both values, the differences between the four enzymes were not dramatic for each antibiotic (less than an order of magnitude), suggesting that all can confer resistance to the antibiotics tested, except aztreonam, which was not converted by any of the enzymes. This is in agreement with the MIC data (Figure 2b and Table 1). The catalytic efficiencies deviated slightly more between enzymes than the turnover numbers. IMP-1 had the highest *k*_cat_/*K*_M_ against benzylpenicillin, cephalothin, cefoxitin, and doripenem, while the double mutant IMP-1-S115T-S119G was the most efficient enzyme against cefotaxime, ceftazidime, and meropenem. Interestingly, IMP-1-S119G was very efficient against both penicillins, ceftazidime, and the carbapenems, while IMP-1-S115T was less efficient against those substrates. Thus, IMP-1-119G seemed to compensate partially for a lower expression level (Figure 3 and Table 2) with higher enzyme activity, resulting in sufficient MIC values to confer resistance (Figure 2b).

## 4. Discussion

The ability of IMP enzymes to confer resistance depends on several factors: they need to be properly expressed, delivered to the periplasm, and processed (cleavage of an 18 amino acid leader sequence [30]); Zn(II) ions need to be incorporated; the enzyme needs to fold into its native conformation; and the completed enzyme needs to bind efficiently to and convert important β-lactam antibiotics. The present study indicated that the four IMP variants investigated here all fulfilled all these requirements. They all conferred resistance to the panel of antibiotics tested except aztreonam, which is known to not be a substrate of IMP-type MBLs [30] (Figure 2b). However, there were some subtle differences: the MICs of the penicillins, third-generation cephalosporins, and the newer carbapenems were higher with IMP-1 and the double mutant IMP-1-S115T-S119G than with the single mutants IMP-1-S115T and IMP-1-S119G. This observation supports the notion that both mutations in combination were more beneficial than either one mutation by itself, which was also consistent with the fact that variants with the combinations S115 and G119 or T115 and S119 have not been isolated. On the phylogenetic tree (Figure 1b), all variants with both mutations were on the same branch together with IMP-46 (T115 and D119). Assuming that these variants evolved from a common ancestor, it appeared that once both mutations were acquired, they proved beneficial and were retained. Therefore, in response to our original question, our data indicated that there was a functional benefit of the two mutations occurring in combination rather than being coincidental. An interesting follow-up question is: What are the roles of both mutations in their combination, and in which order were they likely acquired?

The Role of S115T: Structurally, the serine to threonine mutation is very conservative, as both residues contain a hydroxyl group with the only difference being that a Cβ hydrogen in serine is replaced with a methyl group in threonine. In crystal structures [8,31], the hydroxyl group of S115, which is located at the transition between β strand 5 and loop 7, points to the center of the protein (Figure 1a) and forms a hydrogen bond with D84 in β strand 4. Mutating S115 to threonine with the Swiss PDB Viewer mutate tool [32] retains this hydrogen bond and places the methyl group at the β sheet interface, 3.1 Å from the side chain of F218 in β strand 10 and 4.8 Å and 4.5 Å, respectively, from the side chains of V201 and V203 in β strand 9. Thus, the extra methyl group in T115 seems to contribute to the hydrophobic core of the protein, providing a rationale for the increased thermal stability (3.5 °C higher than that of IMP-1). Hydrophobic core packing is a well-known contributor to the stability of globular proteins [33]. The S115T mutation also led to a relatively high expression level of 95% and therefore almost identical to that of IMP-1. On the other hand, in terms of enzyme activity, IMP-1-S115T is an inferior enzyme, exhibiting the lowest *k*_cat_ against cefotaxime, ceftazidime, meropenem, and doripenem and the lowest *k*_cat_/*K*_M_ against benzylpenicillin, ampicillin, cefoxitin, cefotaxime, ceftazidime, meropenem, and doripenem. In terms of MIC values, this enzyme also conferred the lowest resistance levels to benzylpenicillin and ampicillin and no increased resistance levels to any of the antibiotics tested relative to IMP-1. In other words, there would be no evolutionary benefit in selecting the S115T mutant by itself. The decreased enzyme activity could be the result of a more rigid loop 7. In a mutational study of BcII, flexibility was associated with increased enzyme activity [34].

The Role of S119G: In crystal structures [8,31], the side chain of S119 points toward the solvent and is not engaged in any hydrogen bonds. However, there is an interaction with the aromatic π electron system of F87 (3.0 A) [8] or the side chain of K150a (3.6 A) [31]. These interactions do not seem to prevent mutation to glycine, which is expected to make loop 7 more flexible. The increased flexibility could be responsible for more efficient enzyme catalysis [34] and for decreased thermal stability and expression levels. D119 in IMP-46 could form a salt bridge with K150a. Western blots of *E. coli* DH10B cells expressing the MBLs under the same conditions as in the MIC assays revealed that while IMP-1-S115T and IMP-1-S115T-S119G are expressed at similar levels as IMP-1, the expression level of IMP-1-S119G is about half of that of IMP-1, which can readily explain some of the lower MICs with IMP-1-S119G (cefotaxime, ceftazidime, meropenem, and doripenem). The lower yield of IMP-1-S119G obtained after overexpression and purification is also consistent with a lower expression level. In addition, IMP-1-S119G was the least thermally stable variant, although it was still very stable with a melting temperature of 71.2 °C. Once folded into its native conformation, both Zn(II) ions are bound tightly, so that their binding is maintained after extensive dialysis (Table 2). Perhaps this variant does not fold as efficiently as the others. Codon bias can be excluded as a factor, as IMP-1-S115T-S119G is expressed efficiently with the same codon coding for G119. It cannot be excluded, though, that the particular mRNA encoding IMP-1-S119G folds into a secondary structure that is more quickly degraded or is less efficiently translated. An analysis with mfold [35] did not support that notion. The secondary structures and folding energies of the IMP-1, IMP-1-S115T, and IMP-1-S119G transcripts were identical. Only that of IMP-1-S115T-S119G deviated in secondary structure and folding energy by ~7 kcal/mol (more negative).

In contrast to IMP-1-S119G’s decreased expression level and thermal stability, it exhibited high enzyme activity with higher *k*_cat_ values against ampicillin, cefotaxime, and doripenem and higher *k*_cat_/*K*_M_ values against ampicillin and imipenem than any of the other three enzymes. These could, at least in part, compensate for the decreased expression level and thermal stability.

Combination of S115T and S119G: These observations support the following scenario, which has frequently been observed in the evolution of enzymes: (1) one mutation (in this case S119G) improves enzyme activity, but at a cost, in this case decreased expression level and thermal stability, and (2) a secondary mutation (in this case S115T) compensates for the detrimental effects introduced by the first mutation. For instance, Tomatis et al. showed that mutation G262S in the MBL BcII increased activity, while a second mutation, N70S, led to more enzyme flexibility, which was favorable in that enzyme [34]. Huang and Palzkill have identified the naturally occurring M182T mutation in TEM serine β-lactamases to act as a suppressor of folding or stability defects introduced by other mutations associated with drug resistance [36]. These results were recapitulated in a directed evolution study [37]. Winkler and Bonomo identified stabilizing suppressor mutations in the closely related SHV enzymes [38]. Based on the amino acid changes investigated here, it seems reasonable to assume that the S119G mutation increased the flexibility of loop 7, which may facilitate positioning of Zn(II) ligands for more efficient substrate conversion, albeit decreasing the enzyme’s stability and expression level. The S115T mutation, introducing a hydrophobic group in the protein core, on the other hand, is expected to decrease flexibility and could provide a structural rationale for a stabilizing effect, while maintaining most of the activity gains of S119G. The thermal stability of the double mutant was 2.2 °C higher than and the expression level comparable to (84%) that of IMP-1.

The scenario is also supported by genetic considerations. Both S119G and S115T can be obtained by single nucleotide changes (AGC → GGC for S119G and TCT → ACT for S115T). However, D119 in IMP-46 can only be obtained by a single nucleotide change from G119 (GGC → GAC). The mutation S119D requires two nucleotide changes (AGC → GAC), which would be very unlikely to occur together. Therefore, it seems likely that the G119D mutation was acquired from an already established enzyme harboring both G119 and T115.

## 5. Conclusions

This study took a microbiological, biophysical, and biochemical approach to investigate the roles of mutations S115T and S119G observed in nine out of 80 currently reported IMP variants. The two mutations always occurred in combination, and our data indicated that the combination was more beneficial than either mutation by itself. Specifically, S119G enhanced enzyme activity, but with negative effects on expression level and protein stability. S115T suppressed those defects while maintaining most of S119G’s gains with respect to activity. This study helps understand which factors drive evolution in IMP-type MBLs, knowledge that may prove valuable in the prediction of future mutations and in the design of new antibiotics and MBL inhibitors.

## Figures and Tables

**Figure 1 biomolecules-09-00724-f001:**
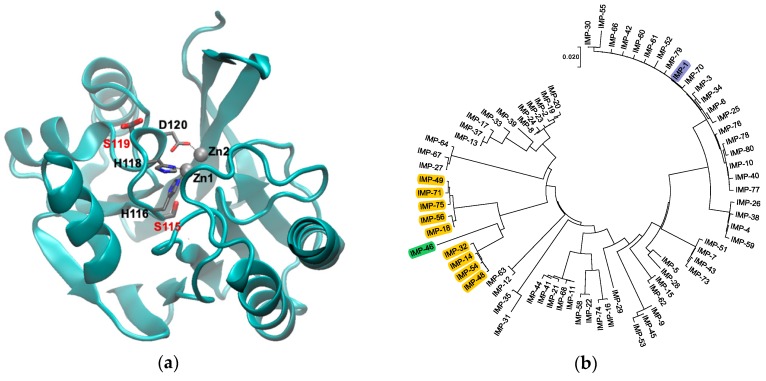
(**a**) Graphical representation of IMP-1 (PDB Code 1DD6 [8]) with Zn(II) shown as gray spheres and Zn(II) ligands in loop 7 as thin sticks. The two residues mutated in this study, S115 and S119, are shown as thick sticks labeled in red. Note that S119 is shown in two alternative conformations as per the crystal structure. The figure was generated with VMD [14] Version 1.9.3. The backbone is shown as a cyan cartoon. Residues are colored by atom: C, gray; N, blue; O, red. (**b**) Phylogenetic tree of the currently known IMP-type variants. Variants containing the S115T and S119G mutations studied here are highlighted in yellow, the IMP-1 reference enzyme in blue, and variant IMP-46 harboring S115T in combination with S119D in green. The tree was generated with MEGA [15] Version 7.

**Figure 2 biomolecules-09-00724-f002:**
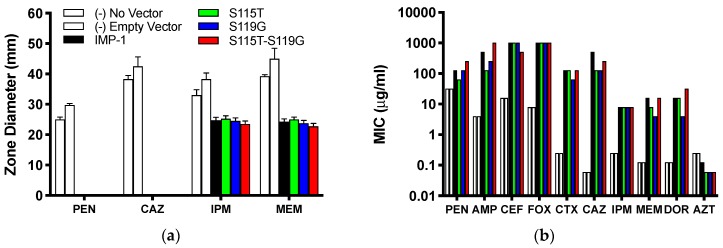
(**a**) Inhibition zone diameters observed in agar disc diffusion assays with DH10B cells expressing the four different enzymes from the pBC SK(+) vector and negative controls. Means ± standard deviations of four replicates are shown. (**b**) MICs of a range of β-lactam antibiotics. Cells, vectors, and color code for the different enzymes and controls are the same as in Panel (**a**). PEN = benzylpenicillin; AMP = ampicillin; CEF = cephalothin; FOX = cefoxitin; CTX = cefotaxime; CAZ = ceftazidime; IPM = imipenem; MEM = meropenem; DOR = doripenem; AZT = aztreonam. Median values of four replicates are reported. The values of AMP with IMP-1-S115T-S119G and FOX with all enzymes are ≥1024 μg/mL, so they could be higher than graphed.

**Figure 3 biomolecules-09-00724-f003:**
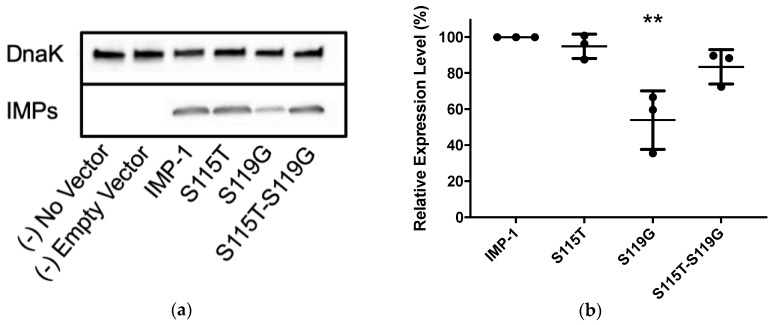
(**a**) Representative Western blot of the four different IMP variants expressed in *E. coli* DH10B cells. DnaK was used as a constitutively expressed marker. The different IMP variants were labeled with a previously reported anti-IMP-1 antibody [22,23]. Exposure times were optimized to obtain good signals without saturation. (**b**) Quantification of three different Western blot experiments. Only the expression level of IMP-1-S119G was significantly lower than that of IMP-1 (*p* < 0.01).

**Figure 4 biomolecules-09-00724-f004:**
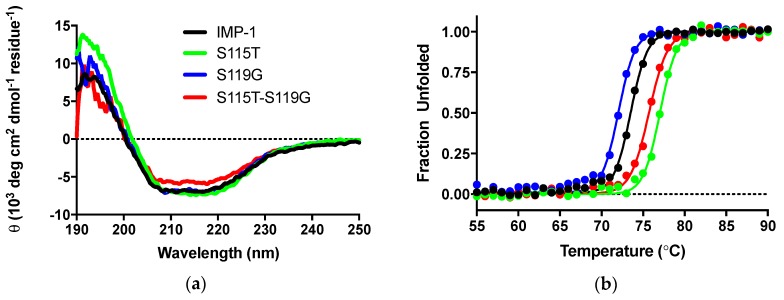
(**a**) CD scans of the four enzymes indicating that the secondary structures are comparable. θ indicates ellipticity. (**b**) Thermal denaturation monitored by CD at 215 nm.

**Figure 5 biomolecules-09-00724-f005:**
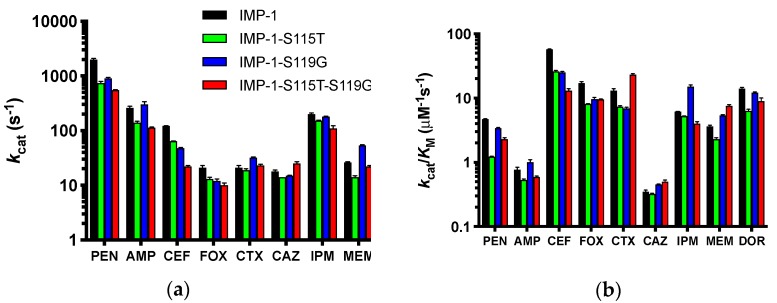
Graphical representation of the kinetic constants of the four enzymes against a panel of antibiotics spanning penicillins, cephalosporins, a cephamycin, and carbapenems. The monobactam aztreonam was not converted and is not included. The abbreviations of antibiotics are as in Figure 2. Columns are colored by enzyme variant as shown in the legend in Panel (a). (**a**) Turnover numbers *k*_cat_. (**b**) Catalytic efficiencies *k*_cat_/*K*_M_. The y-axes are shown in log scale to facilitate comparison of all antibiotics.

**Table 1 biomolecules-09-00724-t001:** MICs (μg/mL) of a range of antibiotics. Medians of four replicates are reported.

β-Lactam	pBCSK(+) *bla*_IMP-1_	pBCSK(+) *bla*_IMP-1-S115T_	pBCSK(+) *bla*_IMP-1-S119G_	pBCSK(+) *bla*_IMP-1-S115T-S119G_	No Vector	Empty pBCSK(+)
Benzylpenicillin	128	64	128	256	32	32
Ampicillin	512	128	256	≥1024	4	4
Cephalothin	1024	1024	1024	512	16	16
Cefoxitin	≥1024	≥1024	≥1024	≥1024	8	8
Cefotaxime	128	128	64	128	8	8
Ceftazidime	512	128	128	256	0.063	0.063
Imipenem	8	8	8	8	0.25	0.25
Meropenem	16	8	4	16	0.125	0.125
Doripenem	16	16	4	32	0.125	0.125
Aztreonam	0.125	0.063	0.063	0.063	0.25	0.25

**Table 2 biomolecules-09-00724-t002:** Purification results and biophysical characteristics of purified IMP variants.

Characteristic\Enzyme	IMP-1	IMP-1-S115T	IMP-1-S119T	IMP-1-S115T-S119G
Relative In-Cell Expression Level (%)	100	95	54	83
Concentration After Purification (μM)	93.7	80.4	47.3	52.9
Yield of Purified Enzyme from 1 L Culture (mg)	5.2	4.2	1.3	3.6
Molecular Mass (measured)	25,112	25,125	25,082	25,096
Molecular Mass (calculated) ^1^	25,112.7	25,126.7	25,082.7	25,096.7
Zinc Content	2.2 ± 0.2	2.1 ± 0.2	2.1 ± 0.2	2.3 ± 0.1
Secondary Structure Elements: ^2^				
α Helix (%)	37	37	37	37
β Sheet (%)	26	26	26	26
Random Coil (%)	38	38	38	38
Melting Temperature (°C) ^3^	73.5 ± 0.1	77.0 ± 0.1	72.1 ± 0.1	75.7 ± 0.1

^1^ The molecular masses were calculated with the Compute pI/Mw program [25]. ^2^ Percentages are as output by the DichroWeb [27,28] program K2D. Due to rounding, they add up to be 101%. ^3^ Melting temperatures ± errors are the result of nonlinear fitting to the normalized data in Prism 7.

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
