# Peer review of "Mutation S115T in IMP-Type Metallo-β-Lactamases Compensates for Decreased Expression Levels Caused by Mutation S119G"

_biomolecules, 2019, doi:10.3390/biom9110724_

Round 1

Reviewer 1 Report

The study investigated the role of two quite common mutations of IMP variants and their co-occurrence. The topics is very interesting and the manuscript well-written and clear. I have no main concerns and the manuscript deserves prompt publication as it is.

Only minor suggestion will follow.

Lines 31-31. "clinical presence" does not sound appropriate. I would rather  ad in the clinical samples at the end of the phrase.

Paragraph at lines 34-36. i would suggest to cite the ongoing studies on MBL inhibitors. Indeed, many molecules are promising and even if they are underway of clinical trials, they should be mentioned. i refers to i.e. cefiderocol and few others.

2.2 line 94 competent cells should read "competent E.coli"

3.2.1 line 232. At my best Knowledge, pBC SK plasmid confer resistance to ampicillin. Please, check the sentence.

3.2.2 In my opinion the MIC findings (supp. materials) should be reported in the main text.

Reviewer 2 Report

The study reported by Oelschlaeger et al. in their manuscript entitled "Mutation S115T in IMP-type metallo-beta-lactamases compensates for decreased expression levels caused by mutation S119G" delves into the potential synergy between two naturally occurring mutations detected in IMPs, which could represent by themselves an evolutionary advantage. Indeed, the authors successfully dissect the contribution that each one of these mutations have both in the stability and catalytic efficiency of the enzyme variants. The "materials and methods" section has been thoroughly crafted and all information required to understand and reproduce any of the reported assays is there. Also, I would like to commend the authors for the attention to detail they have put into writing this manuscript. My only suggestion would be to further analyze the structural implications that both mutations have on the overall improvement of the enzyme, given that both S115 and S119 are, according to the pdb 1DD6, quite far from the active site. Moreover, the fact that S115T is a very conservative mutation, since both aminoacids exhibit a hydroxyl group in their side chain, should have been mentioned because it is probably involved in a stabilizing hydrogen bond network.

Overall, I believe this work presents the necessary interest, merit and scientific soundness to be published in the present form, with just one minor comment: the abbreviation of aztreonam in Fig.2 footnote is missing. 
